# Atomic model of the F$_{420}$-reducing [NiFe] hydrogenase by electron cryo-microscopy using a direct electron detector

Matteo Allegretti[1], Deryck J Mills[1], Greg McMullan[2], Werner Kühlbrandt[1], Janet Vonck[1]*

[1]Department of Structural Biology, Max Planck Institute of Biophysics, Frankfurt, Germany; [2]Medical Research Council Laboratory of Molecular Biology, Cambridge, United Kingdom

**Abstract** The introduction of direct electron detectors with higher detective quantum efficiency and fast read-out marks the beginning of a new era in electron cryo-microscopy. Using the FEI Falcon II direct electron detector in video mode, we have reconstructed a map at 3.36 Å resolution of the 1.2 MDa F$_{420}$-reducing hydrogenase (Frh) from methanogenic archaea from only 320,000 asymmetric units. Videos frames were aligned by a combination of image and particle alignment procedures to overcome the effects of beam-induced motion. The reconstructed density map shows all secondary structure as well as clear side chain densities for most residues. The full coordination of all cofactors in the electron transfer chain (a [NiFe] center, four [4Fe4S] clusters and an FAD) is clearly visible along with a well-defined substrate access channel. From the rigidity of the complex we conclude that catalysis is diffusion-limited and does not depend on protein flexibility or conformational changes.

## Introduction

*For correspondence: janet.vonck@biophys.mpg.de

It has long been recognized that electron cryo-microscopy (cryo-EM) has the potential to solve protein structures at near-atomic resolution (*Henderson, 1995*). Since the first sub-nanometer resolution cryo-EM structures of icosahedral viruses, which have the advantage of multi-MDa size and high symmetry (*Böttcher et al., 1997*; *Conway et al., 1997*), steady advances in instrumentation have yielded a number of virus structures at better than 4 Å resolution (*Grigorieff and Harrison, 2011*). For smaller and less symmetrical structures, progress has been slower, but several complexes have yielded structures at around 6 Å resolution where secondary structure is recognizable and X-ray structures of subcomplexes can be reliably fitted (*Ludtke et al., 2004*; *Armache et al., 2010*; *Gipson et al., 2010*). In a few cases, side chain densities have been resolved (*Cong et al., 2010*; *Ludtke and Baker, 2008*; *Zhang et al., 2010, 2013*; *Mills et al., 2013*). Radiation damage limits the electron dose that can be used in any individual EM image and the resulting low signal-to-noise ratio (SNR) has meant that until recently high-resolution maps have required the equivalent of 10$^6$ images of asymmetric units.

The recent introduction of direct electron detection cameras with much better detective quantum efficiency (DQE) has meant that images having higher SNR can be obtained with the same, or lower, total electron exposure. In addition, these cameras have frame rates which make it possible to collect videos of the particles in the thin film of vitreous water during an exposure, and thus partially correct for the effects of beam-induced particle movement and specimen drift. Using a 70-MDa virus that can be accurately aligned even at short exposures, it was shown that electron irradiation causes random translations and rotations of the particles, which are typically largest in the first few frames (*Brilot et al., 2012*; *Campbell et al., 2012*). In two recent studies, image processing schemes were developed to

**eLife digest** Many microbes rely on enzymes known as hydrogenases to catalyse the metabolic reactions that generate energy. These enzymes cleave hydrogen molecules to release electrons that go on to participate in further reactions. In order to fully understand how hydrogenases and other enzymes work it is necessary to work out their structure at the atomic level.

Last year a technique known as electron cryo-microscopy (cryo-EM) was used to show that Frh—a hydrogenase that is crucial for many different steps in the metabolic process of microbes that produce methane—had a tetrahedral structure. Cryo-EM involves freezing the molecule of interest in a layer of ice to preserve its structure as it is imaged with an electron beam. Unfortunately, the signal-to-noise ratio in each image is low, so researchers must combine many separate images in order to determine the structure of the molecule.

The use of a new type of electron detector can improve the performance of an electron cryo-microscope in several ways. Higher frame rates can be used, which makes it possible to correct for movement of the molecule caused by the electron beam. The new electron detectors are also more efficient, so samples can be exposed to lower doses of electrons, reducing damage to the sample.

Using the new direct electron detectors, Allegretti et al. were able to work out the structure of Frh in greater detail than before. The results confirm that the previously reported structure is correct. Furthermore, several new structural features were seen for the first time, including a previously unseen ion located between two protein subunits. Allegretti et al. also revealed that the structure of Frh is highly rigid, and so the process by which it catalyses reactions involving its substrate, the coenzyme $F_{420}$, does not involve changes in its shape. Instead, the reaction rate depends on the rate at which $F_{420}$ diffuses to the correct position in the enzyme, where the reaction occurs very rapidly.

correct for these beam-induced motions. Using the 700 kDa archaeal 20S proteasome, which is too small to reliably detect and align in images recorded with a short exposure, a protocol was developed to align entire frames or subareas of >2000 × 2000 pixels to each other (*Li et al., 2013*) (The motion correction software is available to download from http://www.nature.com/nmeth/journal/v10/n6/extref/nmeth.2472-S2.zip). For the much larger ribosome (~4 MDa) a statistical video processing approach was developed, which acts on individual particles using a user-defined running average of video frames (*Bai et al., 2013*). Both studies produced maps of unprecedented resolution, 3.3 Å for the D7 proteasome from 126,000 particles and 4.5 Å from 35,000 asymmetric 80S ribosomes.

Recently we determined the structure of the 1.2 MDa Frh complex, the $F_{420}$-dependent hydrogenase from *Methanothermobacter marburgensis*, *ab initio* from cryo-EM data collected on photographic film (*Mills et al., 2013*). Frh is a key enzyme in the metabolism of methanogenic archaea, where the reduction of carbon dioxide to methane involves the oxidation of four molecules of $H_2$ by a number of different hydrogenases. The reduced form of the $F_{420}$ coenzyme, which is the electron donor in several of these steps, is regenerated by Frh (*Thauer et al., 2010*). The central role of Frh in the metabolism of methanogens is reflected in its abundance (~2%) in the soluble cell protein (*Fox et al., 1987*). Frh is a heterotrimeric enzyme composed of the 43 kDa subunit FrhA that contains a [NiFe]-center, the 26 kDa subunit FrhG with three [4Fe4S] clusters, and the 31 kDa iron–sulphur flavoprotein FrhB, which contains the $F_{420}$-binding site and has one [4Fe4S] cluster and an FAD. Our cryo-EM study showed that the Frh complex is a dodecamer with tetrahedral symmetry. The map had ~5 Å resolution which was sufficient to show secondary structure and density for many side chains as well as the cofactors forming the electron transfer chain, making it possible to trace the three protein chains, one of them without a template.

We have now collected a new dataset of Frh as videos on a back-thinned Falcon II detector and reconstructed a cryo-EM using a 'gold standard' refinement approach. The resulting map from 26,000 particles has a resolution of 3.36 Å, which enabled us to refine the model obtained from film data. We can now trace the three proteins in the complex completely, and localize most side chains with confidence. We compare and discuss different processing schemes to deal with the video data.

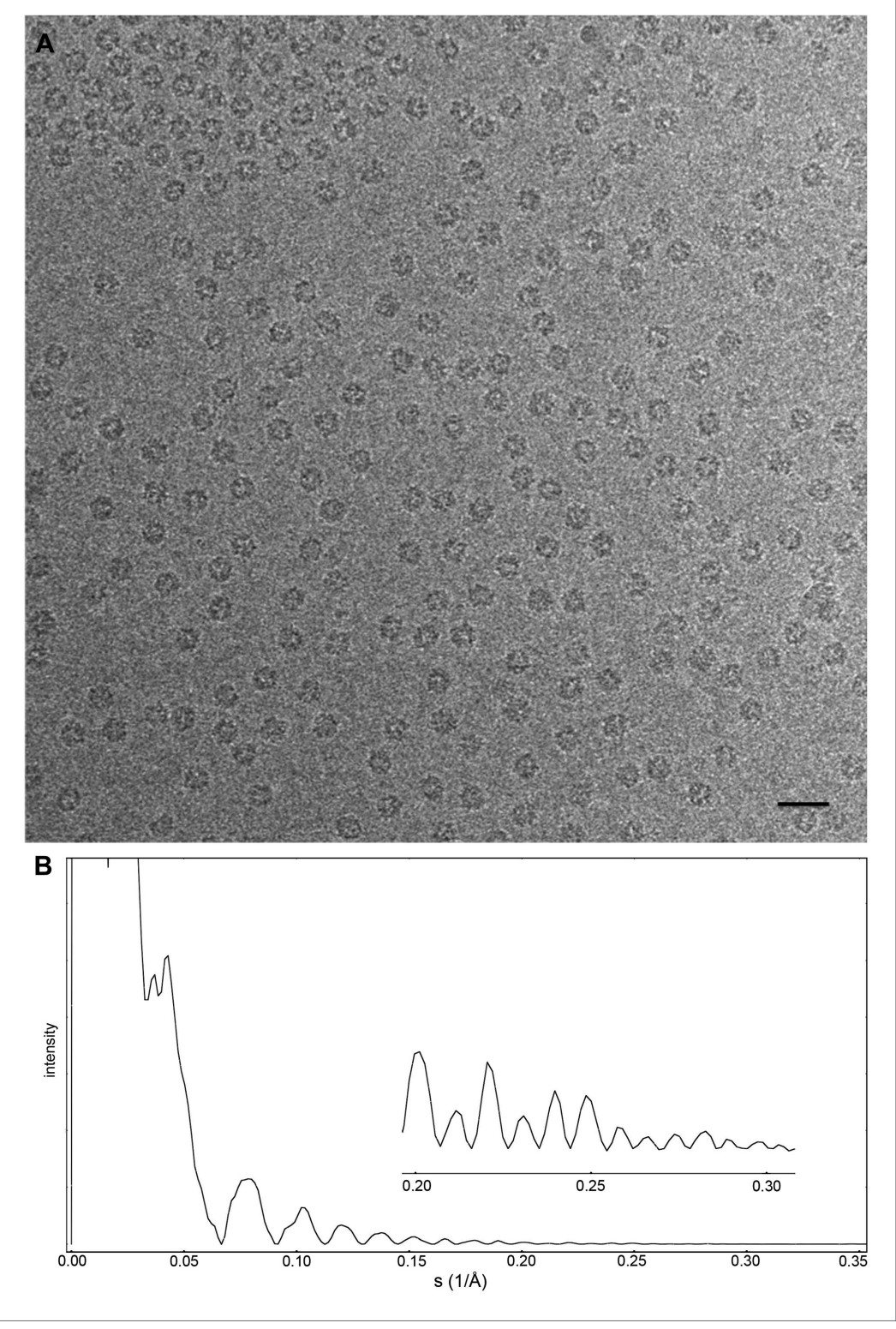

**Figure 1**. Cryo-EM data collection of the Frh complex and CTF correction. (**A**) A typical electron micrograph recorded with the Falcon II camera on the FEI Tecnai Polara operated at 300 kV. The defocus was determined by CTFFIND3 to be 0.9 µm (***Mindell and Grigorieff, 2003***). The particles are clearly visible and easy to box. Scale bar, 25 nm. (**B**) CTF of boxed particles. The inset shows a zoom of the high-resolution range. Thon rings are visible beyond 80% of the Nyquist frequency at 0.3 Å⁻¹.

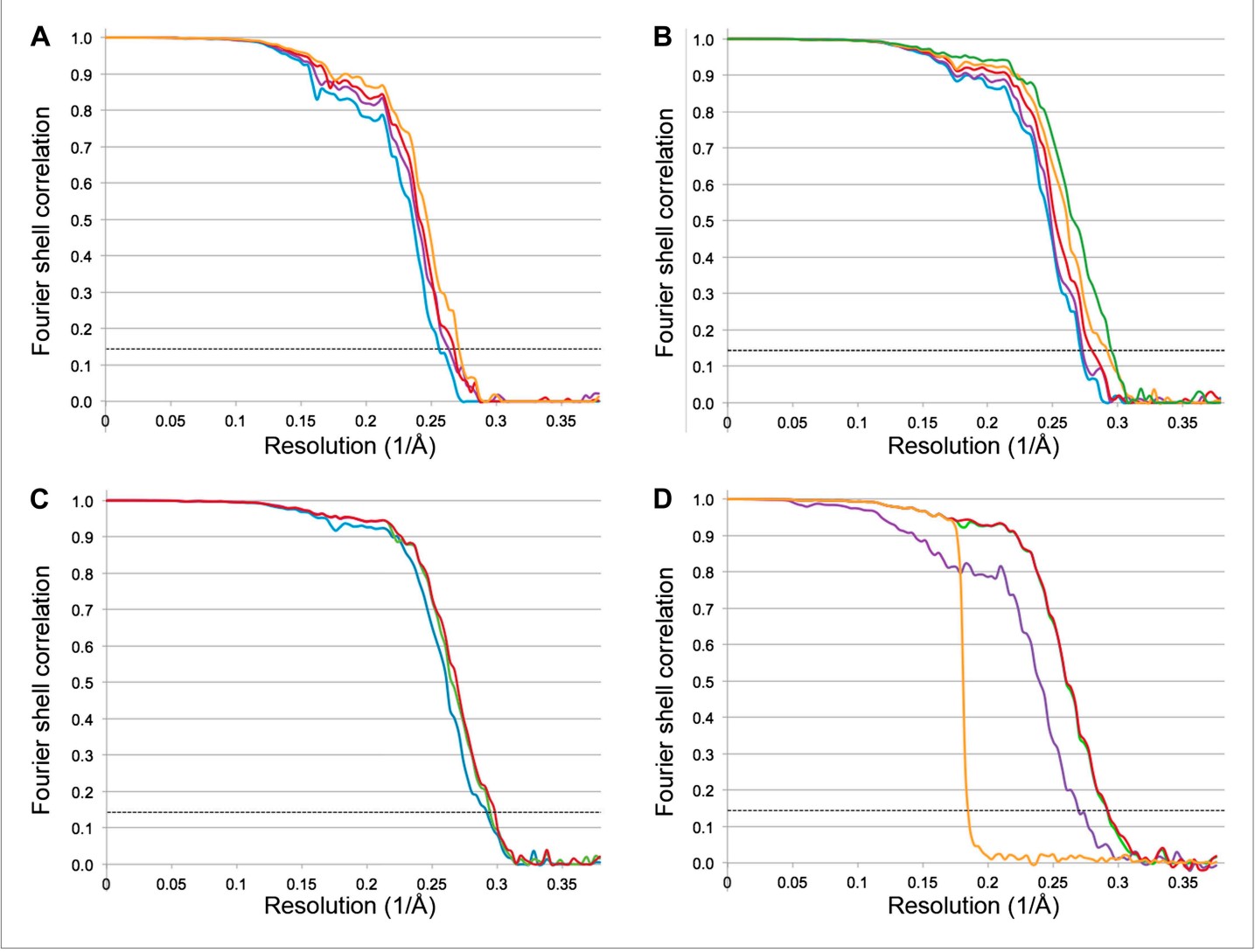

**Figure 2**. Fourier shell correlation (FSC) curves for different refinement strategies. All refinements were performed with the gold standard procedure in RELION (*Scheres and Chen, 2012*). A post-processing procedure (*Chen et al., 2013*) was applied unless otherwise indicated. The dotted line is at FSC 0.143, used to determine the resolution from comparing two independently refined half data sets (*Rosenthal and Henderson, 2003*). (**A**) Comparison of different alignment procedures. Blue, average of 20 unaligned frames. Purple, 20 frames aligned with the statistical video processing procedure (*Bai et al., 2013*). Red, refinement after aligning frames with the area-motion correction software (*Li et al., 2013*). Gold, combination of the latter two alignment procedures. (**B**) Effect of radiation damage. Blue, 20 frames (3.69 Å); purple, 16 frames (3.69 Å); red, 12 frames (3.60 Å); gold, 8 frames (3.43 Å); green, 6 frames (3.39 Å). All curves were obtained by the combination approach described above. (**C**) Data set quality. Particle images of sub-standard quality were omitted from the original data set of 33,590 images, yielding a smaller dataset of 26,635 particles ('Results'). Refinement using 8 frames of the reduced data set (green, 3.39 Å). The improvement is clear in comparison with the full dataset (blue, 3.43 Å). A further improvement (red, 3.36 Å) resulted from using 6 instead of 8 frames. (**D**) Post-processing to determine the resolution and B-factor (*Chen et al., 2013*) of the final map from the 6-frame refinement. The raw unmasked map (purple) indicates a resolution of 3.52 Å; the map masked with a soft mask (red) indicates the final resolution of 3.36 Å. The gold curve shows the FSC for the two half data sets with randomized phases beyond 4.5 Å. Subtraction of the gold curve from the red curve yields the green curve, which indicates the true map resolution corrected for over-aggressive masking. The close correspondence of the red and green curves shows that the used soft mask did not introduce spurious correlation and the true map resolution is 3.36 Å. A B factor of −156 Å² was determined and applied to sharpen the map.

## Results

Images of the Frh complex were collected using the back-thinned FEI Falcon-II in video mode with 18 frames per second. Using videos allows long exposures in which the individual particles and Thon rings are clearly visible, while retaining the radiation-sensitive higher resolution information only present

during the initial exposure. The particles were clearly distinguishable below 1 μm defocus with a 1.5 s exposure (~90 e/Å²) (*Figure 1A*). A dataset of 33,590 particles (403,080 asymmetric units) at a defocus range of 0.8–3.5 μm was collected from 235 images in two sessions on a single grid. Areas of very thin ice were selected that we estimate to be only slightly thicker than the particle diameter of ~175 Å; often similar holes nearby had no particles in the center but a densely packed outer rim, so apparently the thin ice had squeezed the complexes out to the edge of the hole.

The Frh dodecamer was refined from a low-pass filtered model to high resolution using the RELION gold standard refinement procedure. In this protocol, two half datasets are refined completely independently throughout, and after each cycle the new reference volumes are low-pass filtered to the resolution where the Fourier shell correlation (FSC) between the two volumes drops to 0.143, thus preventing overfitting of noise (*Scheres and Chen, 2012*). A first refinement using the sum of the first 20 video frames (discarding the last 4 frames because of likely radiation damage [*Baker and Rubinstein, 2010*]) yielded a map at 3.94 Å resolution (*Figure 2A*) that looked considerably better than our previous map from 84,000 particles on film. Also, the map was virtually noise-free due to the RELION refinement procedure (*Scheres, 2012*). Next we applied the frame alignment software (*Li et al., 2013*) and aligned all 24 frames to each other. In the majority of cases, the first frame showed a large movement in comparison with the others (*Figure 3A*). We refined the sum of the aligned frames with or without the first frame and found that discarding the first frame results in a higher-resolution map (*Figure 3B*). As found also by others (*Li et al., 2013*), the correction cannot deal with the fast motion of the particles during the first frame. On this basis we discarded the first frame of every video for all subsequent refinements. The refinement of the 20 aligned frames resulted in a 3.74 Å map (*Figure 2A*). In another refinement approach, we used the statistical alignment procedure (*Bai et al., 2013*) implemented in RELION (*Scheres, 2012*). This method follows the movement of the particles within the videos (particle-based method) using a running average of frames. This procedure also improved the resolution relative to the unaligned particles in a very similar way to the whole-frame alignment (*Figure 2A*); the best resolution was obtained when 5 frames were averaged. In yet another approach we combined the two alignment procedures (whole-frame and particle-based). The resolution curve of this refinement gave a better FSC than the others over all frequencies and the resulting reconstruction had a resolution of 3.69 Å (*Figure 2A*), indicating that for the intermediate-sized particle investigated here the combination of the two alignment procedures works best.

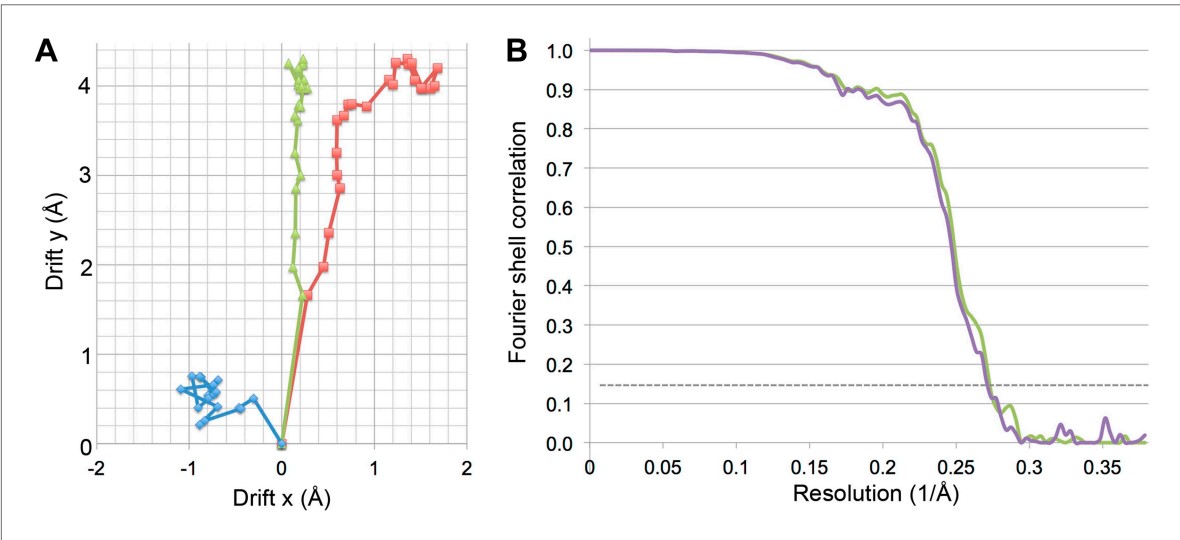

**Figure 3**. Specimen movement as detected by recording images in video mode. The motion correction software indicates a large movement at the beginning of the exposure. (**A**) Video frame alignment for three separate micrographs. Each spot represents one frame. The drift plots show that the movement between the first and second frame is considerably higher than in subsequent frames, although some micrographs (red and green) indicate much higher drift than others (blue). (**B**) FSC curves of a refinement with and without the first video frame. The violet curve represents a refinement of frames 1–17, the green curve frames 2–17. Although at FSC 0.143 the resolution is the same for for both maps (3.69 Å), the green curve without the first frame clearly shows a higher FSC in the whole resolution range.

We had already discarded the last 4 frames of the 24-frame videos, because of likely radiation damage. We then continued to assess the effect of radiation damage on the quality of the map by successively reducing the number of frames. Every omitted frame improved the map resolution until only 6 frames were averaged. This gave the best resolution of 3.39 Å, compared to 3.74 Å for all 20 frames (*Figure 2B*). Reducing the number of averaged frames further to 4 or 5 made the resolution worse.

We attempted to improve the resolution further by adding more data. A third microscope session on a different grid yielded another 15,000 particles. Adding them, however, reduced the map resolution and these images were therefore not included in further steps. We assume the data quality was worse because of the thicker ice on this grid, which had made it necessary to use a higher defocus, never less than 1.2 μm, to distinguish the particles. We therefore decided to discard suboptimal images from the first dataset to improve the map. We removed the images with high defocus (>2.5 μm) and those that were of poor quality as judged by the visibility and symmetry of the Thon rings. This reduced the dataset by 20% from 33,590 to 26,635 particles (319,620 asymmetric units). Refinement of this smaller dataset under otherwise identical conditions resulted in maps with better resolution than the full dataset, and the best map, with 6 frames averaged, extended to 3.36 Å resolution (*Figures 2C and 4*). The accumulated dose for this data was ~24 e⁻/Å². We tested the local resolution of the map with the program ResMap (*Kucukelbir et al., 2014*) to identify possible flexible protein regions. The local resolution map was featureless (not shown), indicating that the dodecameric Frh complex is completely rigid.

To assess the effects of radiation damage further, we also created maps of the last 6 and the middle 6 frames of the 20-frame videos of the reduced dataset, averaging particles that were in effect

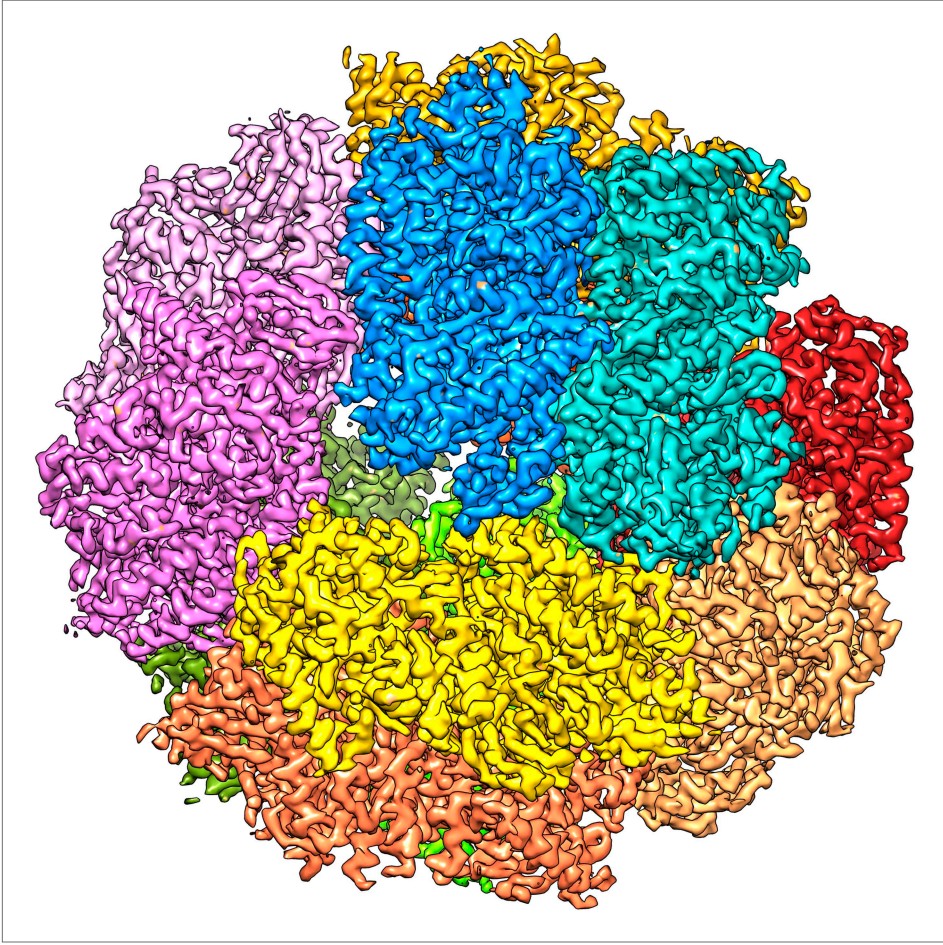

**Figure 4**. The 3.36 Å map with each of the 12 heterotrimers in a different colour.

pre-irradiated with ~24 and 49 e⁻/Å². This resulted in maps of 3.94 and 4.16 Å resolution, respectively, which was clearly reflected in the poorer visibility of side chain densities (*Figure 5*).

We fitted the model of the FrhABG heterotrimer based on the model built into the ~5 Å map obtained previously from film data (*Mills et al., 2013*). The overall features of the 3.36 Å map were similar, but the higher resolution of the new map and the low noise level made the interpretation of the electron densities unambiguous (*Figure 6*; *Video 1*). Alpha helices (*Figure 7*) and beta sheets (*Figure 8*) were easily recognizable and the large majority of side chains had good density (*Figure 9*; *Video 2*). Most of the protein could be fitted to the map using the 'real space refine' feature in Coot, which had not been possible for the lower resolution map from film images. The FSC between this map and a map calculated from the fitted model is 0.5 at 3.56 Å resolution, close to the resolution of the final map, confirming the correctness of the fit (*Figure 10*).

All three subunits were traced completely except for a few N- or C-terminal residues, accounting for 893 out of 903 amino acids (*Figure 11*). We found that all three proteins had been traced essentially correctly in the lower-resolution film map (*Video 3*). The largest differences between the two models were found in surface loops that had been difficult to trace in the film map. The only unresolved part in the previous model involved a 12-residue stretch in FrhG (188–199), for which no density was found and which was thought to form a flexible surface loop. In the new map, two nearby density features, modelled as extended loops before, were clearly recognizable as alpha-helices (194–201 and 208–215) that accommodate the missing residues. The resulting shift of residues brings the cysteine residues C206 and C208, which are conserved in the FrhG family (*Figure 11B*), near a strong density on the dimer axis between two FrhG subunits. The cysteines of the dimer partner are just a few Å away, and the density in between the four cysteine side chains is strongly suggestive of a coordinated ion (*Figure 12*). Although the cysteine side chains are not completely resolved, their tetrahedral arrangement suggests that it is a zinc ion (*Dokmanić et al., 2008*; *Harding et al., 2010*; *Rulíšek and Vondrášek, 1998*; *Zheng et al., 2008*).

The FAD cofactor in FrhB has continuous density in the map, and its folded conformation with the adenine and isoalloxazine moieties in close proximity (*Mills et al., 2013*) is confirmed (*Figure 13*; *Video 4*). As noted before (*Mills et al., 2013*), the FAD is completely surrounded by conserved residues, including loops 23–30, 71–77, and 131–138 (which includes Cys134, a ligand of the nearby FeS cluster i.e., the electron donor to FAD) (*Figure 11C*). The binding pocket is well-defined in the new map, with the pyrophosphate moiety at the N-terminal end of helix 27–39 and the conserved 24QDGG as an extended chain around it (*Figure 13A*). There is no density for the substrate, $F_{420}$, but the FAD is accessible from the surface through a ~5 Å gap between the main body of FrhB and a domain formed by residues 128–188 containing an alpha helix and a three-stranded beta sheet (*Video 5*). This putative substrate access channel is, like the FAD itself, completely lined by conserved residues (*Video 6*). The model for the FrhABG trimer and the dodecamer is shown in *Figure 14*.

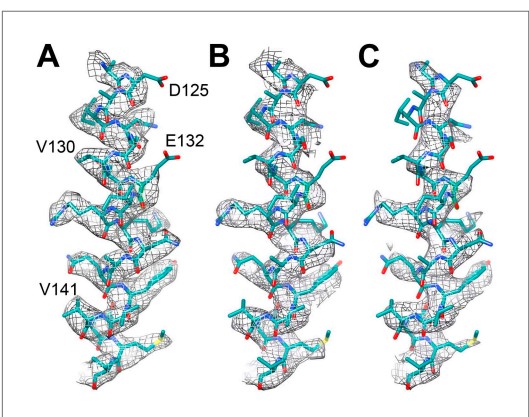

**Figure 5**. Effect of radiation damage. Helix 124–144 of FrhA in the map calculated from (**A**) video frames 1–6 (3.36 Å), (**B**) frames 8–13, pre-irradiated by ~24 e/Å² (3.94 Å), and (**C**) frames 15–20, pre-irradiated by ~49 e/Å² (4.16 Å). Note that side chain density is lacking for Asp125 and Glu132 already in the first map.

## Discussion

Cryo-EM has long had the potential of reaching near-atomic resolution, but until very recently this has only been achieved for very large, highly symmetrical icosahedral viruses. The fundamental problem is the inevitable radiation damage to the biological sample caused by the electron beam. Reaching high resolution requires low exposures to maximize the SNR of rapidly destroyed high-resolution information. The resulting images are noisy, which makes it both difficult to detect particles and to determine the defocus parameters accurately. The high frame rate of direct electron detectors allows data collection in video mode, which removes these restrictions and allows beam-induced motion of particles to be identified and partially corrected for. Just as importantly, direct

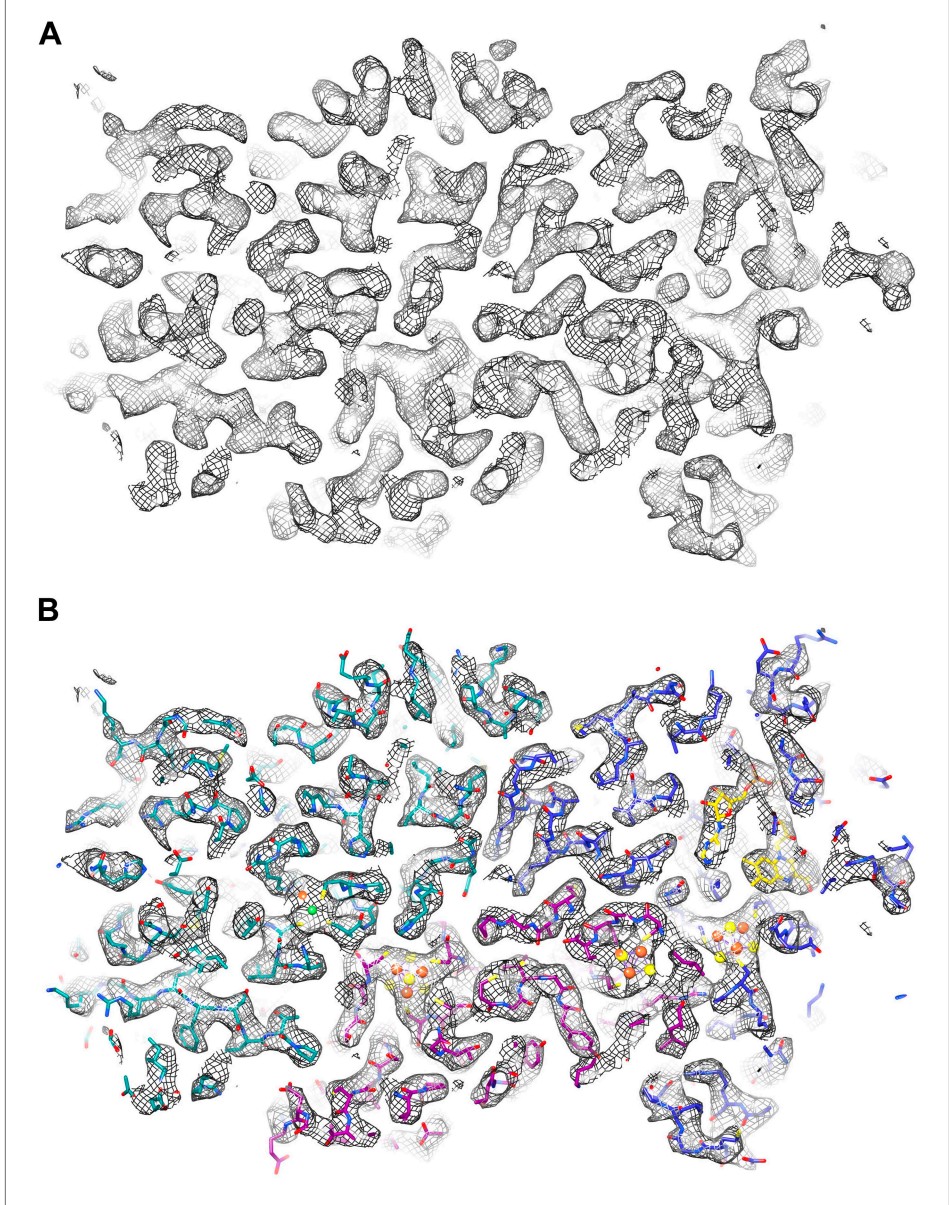

**Figure 6**. EM map of Frh at 3.36 Å resolution. (**A**) Slice through an FrhABG heterotrimer at the level of the electron transfer chain. (**B**) The same slice with atomic model. In this and other figures the carbons of FrhA are green, FrhG magenta, and FrhB blue. Green and orange spheres indicate the [NiFe] center in FrhA; the [4Fe4S] clusters in FrhG and FrhB are shown as orange and yellow spheres. The FAD in FrhB is shown as a stick model with yellow carbons.

electron detectors offer increased detective quantum efficiency at high resolution and so while individual images are still noisy they contain more information. The combination of better images and improved data processing is now leading to better reconstructions which in turn allows more information to be extracted from the images due to better alignment of individual particles.

The advantages are illustrated in recently published studies of two very different specimens, the 4 MDa asymmetric *Saccharomyces cerevisiae* ribosome and the 700 kDa D7-symmetric proteasome. These studies used two different detectors, the FEI Falcon II and the Gatan K2, respectively, and employed different strategies to align the video frames (*Bai et al., 2013*; *Li et al., 2013*). We have now reconstructed the 1.2 MDa tetrahedral Frh complex from ~26,000 particles collected on the Falcon II, and determined the structure of the three proteins in the complex at high resolution.

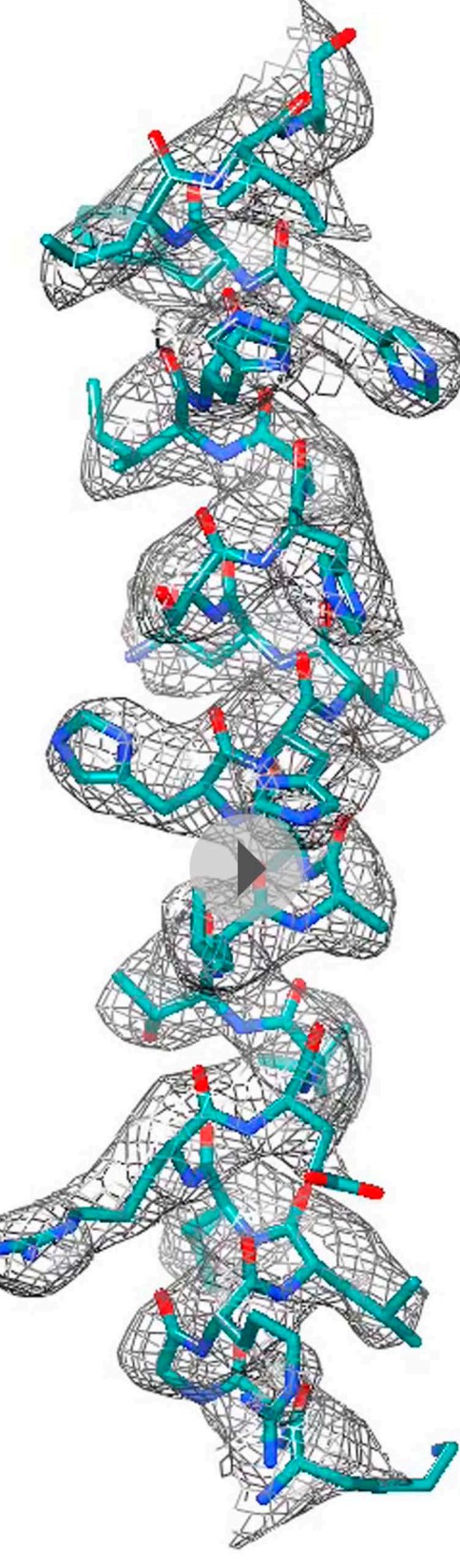

**Video 1**. A slice through the map and model of an FrhABG heterotrimer.

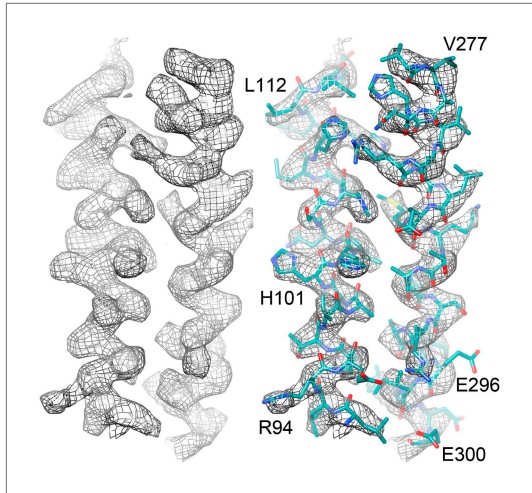

**Figure 7**. Two helices of the 4-helix bundle in FrhA (Leu92-Ala114 and Val276-Glu300) without and with the model. Note the absence of side chain density for glutamate and aspartate side chains.

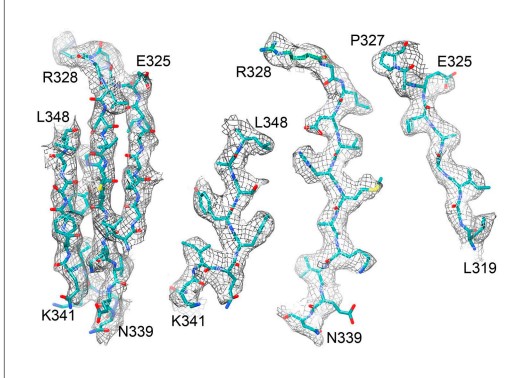

**Figure 8**. Beta sheet 319-348 of FrhA. (**A**) Top view, (**B–D**) side views of individual strands 341–348, 328–339 and 319–327, rotated by 90° relative to (**A**).

The Falcon II data quality is far superior to that obtained with photographic film. With film we obtained a 5 Å map from 84,000 Frh particles (*Mills et al., 2013*), but with Falcon II images of the same sample and the same electron microscope we were able to achieve a map with better than 4 Å resolution from 70% fewer particles. The higher DQE of the detector makes it possible to reliably detect particles at very low defocus. This is illustrated by *Figure 1A*, showing a micrograph taken at 900 nm defocus in which the particles are clearly visible, whereas with film we were not able to detect the particles at less than 1500 nm defocus.

The high-resolution signal for the low-defocus images is very much better than that for images

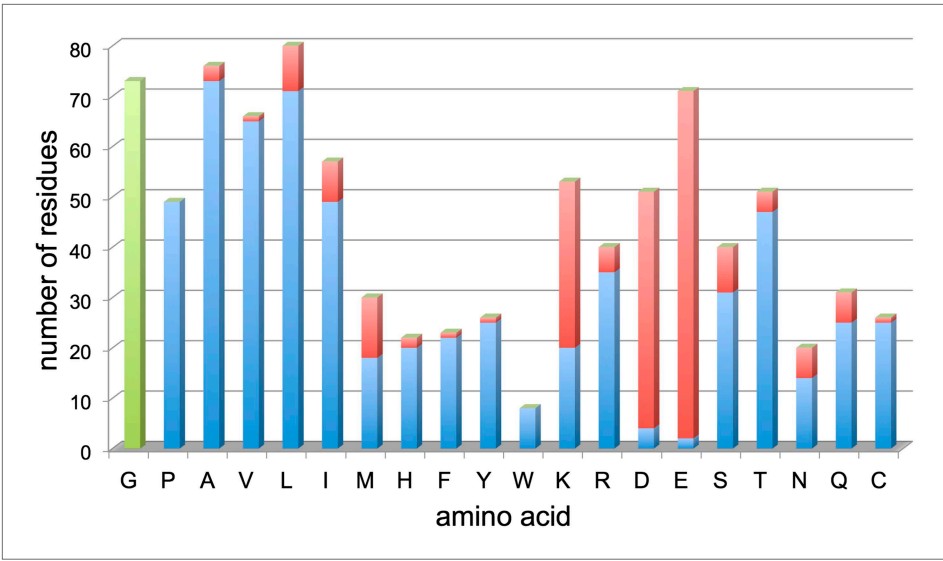

**Figure 9**. Resolved amino acid side chains in the 3.36 Å map. The histogram shows the number of residues on the y-axis and the amino acid (one letter code) on the x-axis. Blue bars indicate fully resolved side chains, red bars indicate side chains without or with ambiguous map density. Glycine residues (no side chain) are indicated in green for completeness. Negatively charged side chains of aspartate (D) and glutamate (E) are almost all missing. In contrast, side chains of hydrophobic residues like valine (V), leucine (L), isoleucine (I), phenylalanine (F), tyrosine (Y), and tryptophan (W), are nearly all visible. Excluding glycine, aspartate and glutamate, 86% of side chains are well visible. Unresolved side chains other than aspartate and glutamate are mostly located on the surface.

recorded at higher defocus, which was a critical factor in attaining the final map. This is illustrated by the fact that adding more particles recorded at higher defocus (from a grid with thicker ice) actually degraded the overall resolution, and that omitting particles with more than 2500 nm defocus, which did not contribute to the signal at high resolution, improved the map (*Figure 2C*).

A decisive advantage of direct electron detectors over film is the possibility of collecting video data. We found that two issues are important here: aligning the video frames to reduce the effects of beam-induced movement, and selecting the frame sequence used in the final map reconstruction to eliminate data affected by radiation damage. We used two approaches for video alignment. The first was the procedure developed by *Li et al. (2013)* for aligning whole frames (or large fractions of frames) independent of the visibility of the particles of interest in the individual frames. This was developed for the 700 kDa 20S proteasome and is most useful for relatively small protein complexes that are hard to detect on images recorded at low electron exposure. The second method is the statistical alignment procedure developed by *Bai et al. (2013)*, which works on the particle level on user-defined running averages of substacks and requires the particles to be visible on the substacks. This was developed for ribosomes, which for a given dose are easier to detect due to their large size (several MDa) and high RNA content. The 1.2 MDa Frh complex, which contains 48 [4Fe4S] clusters and several other metal ions, is intermediate in density and size between the proteasome and the ribosome. For this complex we found that both video alignment methods gave similar improvements in final map resolution compared to unaligned frames (*Figure 2A*). However, the combination of the two methods, running the statistical particle alignment on pre-aligned stacks, yielded another improvement of similar magnitude. We used a

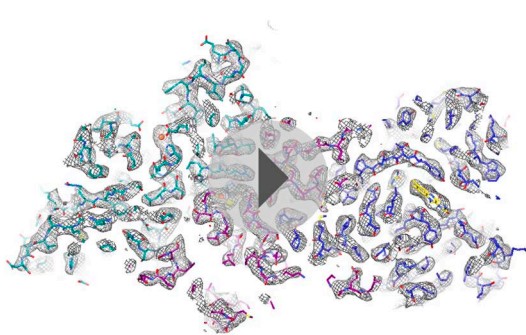

**Video 2**. The quality of the 3.36 Å map shown for alpha-helix 88–114 of FrhA.

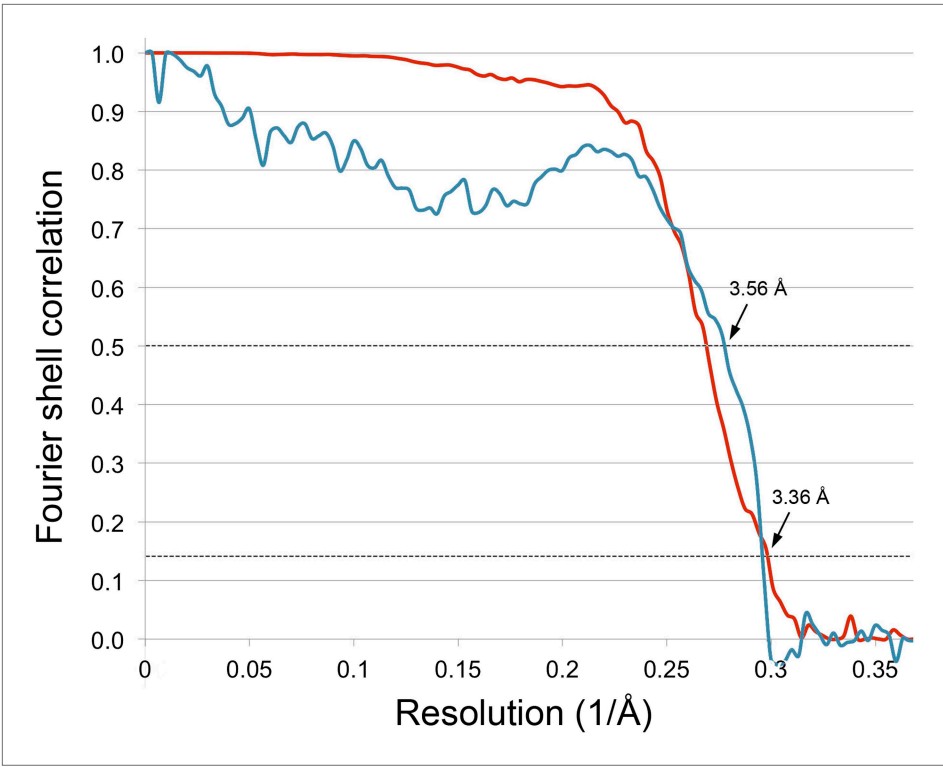

**Figure 10**. Correlation between map and model. The blue line shows the FSC between the final cryo-EM map and a map calculated from the fitted model; the red line is the FSC between maps from independent halves of the data. Dotted lines indicate the 0.5 FSC criterion for the map/model comparison and 0.143 for the half datasets. The cryo-EM map was filtered to 3.36 Å, causing the map-to-model correlation to drop to 0 at that resolution.

running average of 5 frames for the particle-based alignment, so it is not unexpected that a pre-alignment of the frames was advantageous.

Collecting data in video mode also allows elimination of sub-optimal frames from the reconstruction. As was found in earlier studies (*Brilot et al., 2012*; *Campbell et al., 2012*; *Li et al., 2013*), beam-induced movement was much more noticeable in the first frame than in subsequent frames (*Figure 3A*). This large movement during the first fraction of the exposure would blur the first frame, and by omitting it, we obtained an improvement in resolution (*Figure 3B*). The later frames are increasingly affected by radiation damage. Reducing the number of frames used in the reconstruction to only 6 (out of the total 24) produced the best map (*Figure 2B*). The accumulated dose in these frames was ~24 e/Å$^2$, very similar to the accumulated dose (~21 e/Å$^2$) used for the 3.3 Å proteasome map (*Li et al., 2013*). A 4.16 Å reconstruction after a refinement of only the last 6 frames of the 20-frame videos still showed partial side chains and clear helices (*Figure 5C*). In case of smaller particles than the 1.2 MDa Frh complex, it would be advantageous to record videos over longer periods with higher cumulative doses, so that the particles are more visible and easier to align. The later, radiation-damaged frames can then be omitted from the reconstruction if necessary, as suggested before (*Bai et al., 2013*). Thus, direct electron detectors provide a way to optimize a posteriori the electron exposure of the data used for reconstruction.

Our 3.36 Å resolution map made it possible to determine the structure of the Frh complex unambiguously. All elements of secondary structure are obvious, and side chain density is clearly visible for most residues in the protein interior. Side chains on the surface are in general not seen, probably due to their flexibility in the aqueous solvent. The only conspicuous exceptions are the side chains of glutamates and aspartates, which are almost all absent (*Figure 9*), even in maps reconstructed from the earliest, least damaged frames (*Figure 5A*). An absence of aspartate and glutamate side chain densities can also be observed in other high-resolution cryo-EM maps, including the electron crystallography maps of LHC-II (*Kühlbrandt et al., 1994*) and bacteriorhodopsin (*Kimura et al., 1997*)

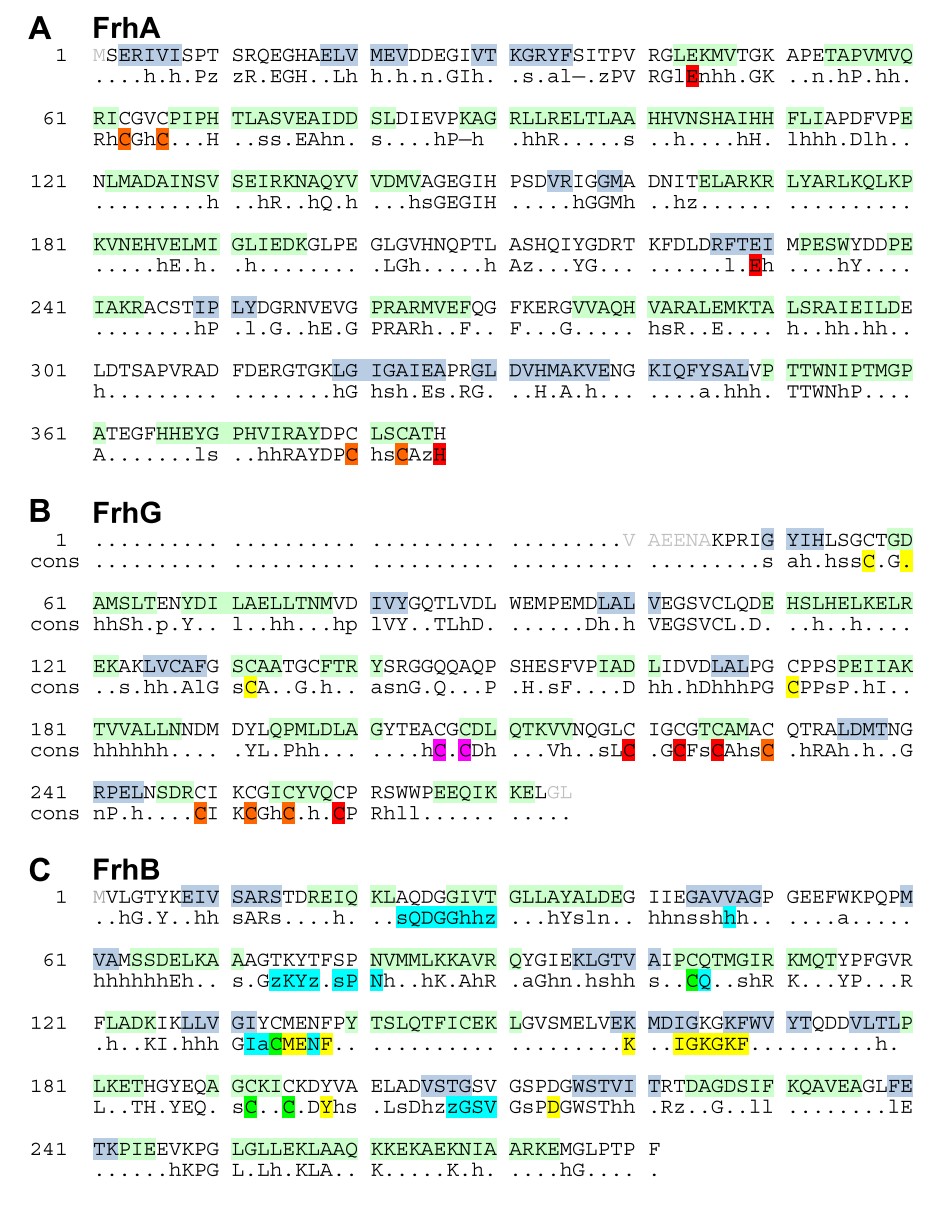

**Figure 11**. Polypeptide sequence and secondary structure. α-helices are highlighted in green, β-strands in blue. The second line shows a consensus sequence of the protein families (*Mills et al., 2013*) with fully conserved amino acids in capitals and partly conserved residues in lower case (h: hydrophobic; s: small [GAS]; l: large [LIFYHW]; a: aromatic [FYWH]; z: T or S; n: negative, D or E; p: positive, R or K). (**A**) FrhA. In the consensus sequence, the [NiFe] ligands are highlighted in orange and the ligands of the third ion in red. (**B**) FrhG. Ligands of the proximal, medial, and distal [4Fe4S] cluster are shown in yellow, orange, and red, respectively. The cysteines coordinating a putative zinc ion on the FrhG dimer interface are highlighted in magenta. (**C**) FrhB. The residues for coordination of the iron–sulphur cluster and FAD are highlighted in green and cyan, respectively, and residues lining the $F_{420}$ access channel in yellow.

and the recent 3.3-Å single particle map of the 20S proteasome (*Li et al., 2013*). In X-ray crystallography it was noted that carboxylate side chains have higher B-factors after extended exposure to intense synchrotron radiation, possibly due to decarboxylation (*Weik et al., 2000*). A quantification of synchrotron radiation damage on carboxylate side chains (*Fioravanti et al., 2006*) shows that this damage already occurs at radiation doses equivalent to less than 1 e/Å² (*Henderson, 1990*), a much lower dose than is feasible in cryo-EM. This implies that the decarboxylation would have occurred well within the

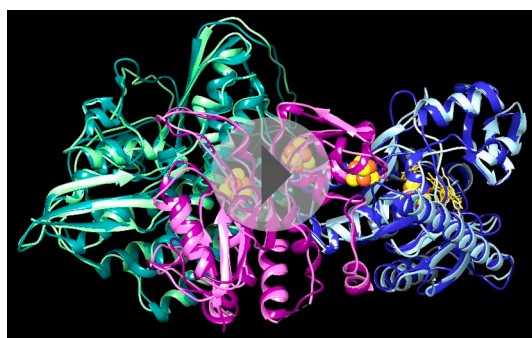

**Video 3**. Comparison of the Frh models based on the film map (*Mills et al., 2013*) and the Falcon II map from the present study. Light colours: film, dark colours: Falcon II. FrhA is shown in green, FrhG magenta, FrhB blue.

first frame of our video data set and aspartate and glutamate side chains would in general not be visible in cryo-EM maps. However, not all are absent. Among the few visible carboxylate side chains in Frh, two are involved in ligand binding: FrhG Asp60 in the coordination of the proximal FeS cluster, and FrhA Glu44 in liganding a possible Fe or Mg ion (*Mills et al., 2013*; *Figure 15*). In X-ray crystallography it is also found that not all glutamate and aspartate side chains are equally susceptible to radiation damage, but the dependence on the structural and chemical environment is not well understood (*Fioravanti et al., 2006*). In the 3-Å bacteriorhodopsin EM map, the side chains of Asp85 and Asp212, which from spectroscopy are known to be deprotonated, were invisible, whereas the protonated side chains of Asp96 and Asp115 had good density (*Kimura et al., 1997*). Active sites are often found to contain the most radiation-sensitive residues in X-ray crystallography, but the protonation state was not found to be essential, and a relation to pKa was also not observed (*Fioravanti et al., 2006*). As more high-resolution cryo-EM maps become available, more data on the relative radiation sensitivity of side chains in different environments may give new insights in protein structure and dynamics.

The structure of Frh was traced ab initio in our previous cryo-EM map from film data, which had a nominal overall resolution of 5.5 Å as determined by gold-standard FSC, but a significantly higher resolution of ~4.5 Å in well-defined regions of the map (*Mills et al., 2013*). There were many indications that our original Frh structure was essentially correct, for example well-resolved hydrophobic side chains in the protein interior and the observation of secondary structure in regions where it was expected from prediction programs. It was nevertheless gratifying to see how well the model fits the new, higher-resolution map. Note that it would hardly be possible to trace an electron density map obtained

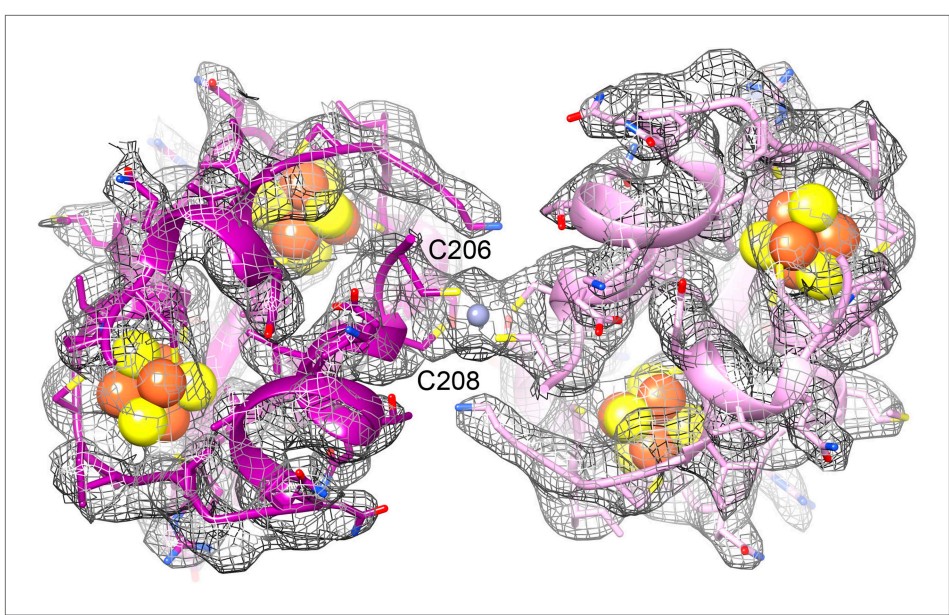

**Figure 12**. The ferredoxin domains (residue 206-260) of an FrhG dimer containing the medial and distal FeS clusters. The two protomers are shown in shades of purple. A high density on the dimer axis between two copies of Cys206 and Cys208 is interpreted as an ion (grey sphere), most likely $Zn^{2+}$. The ion is ~9 Å away from the medial FeS cluster (top left and bottom right).

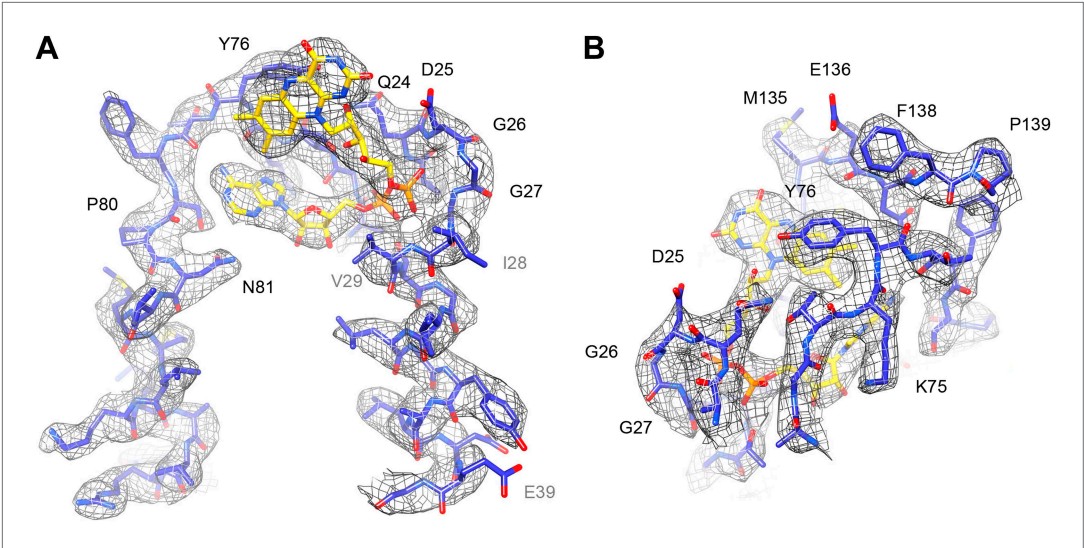

**Figure 13**. FAD cofactor in FrhB with part of its binding pocket. Conserved residues are labelled in black. Other residues mentioned in the text are grey. (**A**) The phosphate moiety (orange) sits in a pocket formed by A23–T30 at the C-terminal end of helix 28–39. The adenine moiety (left) is coordinated by the loop A72–N81. (**B**) In this view, the loop I132–F138 surrounding the isoalloxazine ring can be seen as well as the highly conserved loop G73–T77.

by X-ray crystallography at a nominal resolution of 5.5 Å. This clear difference reflects the quality of the phase information, which is conserved in the phase contrast electron micrographs and determined directly by image processing, whereas in X-ray crystallography it is determined indirectly by isomorphous replacement or anomalous scattering.

Although the overall chain trace is the same, in detail the model based on the 3.36 Å map is of course much improved. The function of Frh is the hydrogenation of $F_{420}$ by molecular hydrogen. Electrons are extracted from hydrogen by the [NiFe] cluster in FrhA, and transferred via the three [4Fe4S] clusters in FrhG and another one in FrhB to FAD and from there to the substrate. The electron transfer chain is clearly recognizable. The [NiFe] cluster has a high density (***Figure 15B***) and is coordinated by conserved residues, as in the large subunits of other [NiFe] hydrogenases (***Figure 11A***). Nearby is another ion, also conserved, coordinated among others by the C-terminal histidine of FrhA, His386 (***Figure 15B***). The four FeS clusters were easily localized in the film map because of their high density; moreover, they are arranged in a chain with distances of ~10 Å (***Mills et al., 2013***). In the Falcon II map, the clusters are not just featureless blobs, but their tetrahedral coordination (mostly by cysteine residues, or, in one case, an aspartate) can be clearly seen (***Figure 6***, ***Figure 15A***). Thus their orientation can be modelled correctly, with the irons facing the sulphurs of the cysteine side chains. One unexpected feature found in the new map was the density for an ion on the dimer interface between two FrhG subunits (***Figure 12***). This density feature was actually visible in the film map, but since the flanking conserved cysteines from the two FrhG subunits were not recognized, its interpretation as an ion was not obvious. The function of this ion is most likely to stabilize the dodecameric complex, but its location between

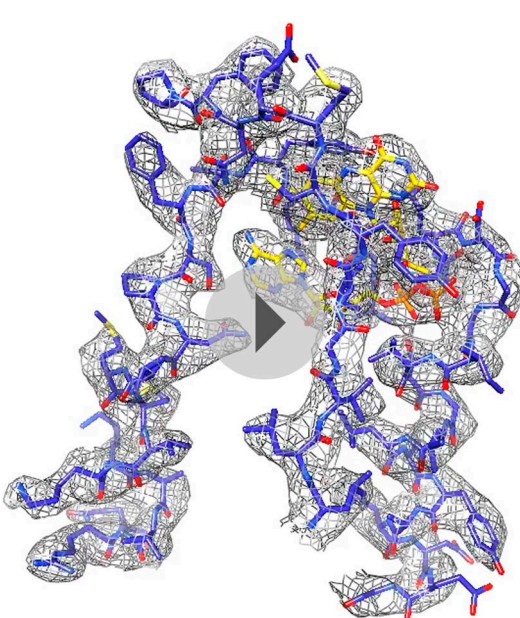

**Video 4**. FAD and its binding pocket.

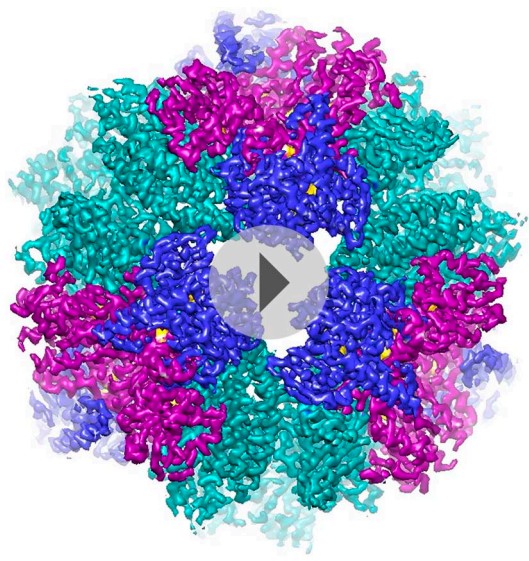

**Video 5**. The 3.36 Å complex map with trimer colours, sliced and rotating to see FAD coloured with FrhA green, FrhG magenta, FrhB blue, and the ligands gold. The substrate access channel from the surface to FAD is clearly visible between two domains of FrhB.

the two medial FeS clusters at a distance of only 9 Å, which is less than the distance of ~10 Å between adjacent FeS clusters in the electron transfer chains, means that a role in electron transfer cannot be ruled out. The final part of the electron transfer chain, the FAD in FrhB, which is close to the last FeS cluster, has unambiguous, continuous density in the Falcon II map (*Figure 13*; *Video 4*). FAD acts as a one-electron/two-electron redox switch between the FeS clusters and the $F_{420}$ substrate (*Fox et al., 1987*; *Alex et al., 1990*; *Thauer et al., 2010*). This means that the substrate needs to approach the FAD to within van der Waals distance to enable hydride transfer (*Ceh et al., 2009*). In our previous study, we collected data from Frh with and without the substrate. In the map obtained in the presence of substrate we had identified a density near the FAD isoalloxazine ring that we interpreted as part of $F_{420}$ at low occupancy (*Mills et al., 2013*). For the new Falcon-II map, we used an Frh sample with a large excess of $F_{420}$, but the $F_{420}$ binding pocket is clearly empty. As the same sample was used in both studies, we conclude that the density in the film map must have been due to noise. An access channel for the substrate can easily be recognized (*Video 5*) and its significance is confirmed by the high conservation of the amino acid residues lining the channel (*Video 6*). There is no indication of even a low occupancy of substrate or of flexibility in this map region. A map of local resolution (*Kucukelbir et al., 2014*) confirmed that the protein region around this pocket is rigid. This suggests that conformational changes do not play a role for the access of $F_{420}$ to the hydride donor, consistent with the finding that $F_{420}$ reduction by Frh is very rapid and most likely diffusion-limited (*Livingston et al., 1987*).

The advent of direct electron detection cameras opens new horizons for cryo-EM. It is now possible to obtain higher-resolution maps with many fewer images. Using the video data collection mode, the optimal data in terms of SNR or radiation damage can be extracted a posteriori, and frame alignment schemes can be fine-tuned to the object under investigation.

## Materials and methods

### Data collection

The purification of the Frh complex from *Methanothermobacter marburgensis* and the preparation of the grids for the electron microscope data collection were performed as described (*Mills et al., 2013*). Briefly, 3 µl of a 0.7 mg/ml Frh sample in the presence of 10 mM $F_{420}$ was applied to freshly glow discharged Quantifoil R1/4 holey carbon grids (Quantifoil Micro Tools, Jena, Germany). The grids were blotted in an FEI Vitrobot plunge-freezer. Data was collected on an FEI Tecnai Polara operating at 300 kV, using a back-thinned FEI Falcon II direct electron detector. The microscope was carefully aligned as previously described (*Mills et al., 2013*) and the Falcon II camera was calibrated at the desired nominal magnification of 78,000×. The

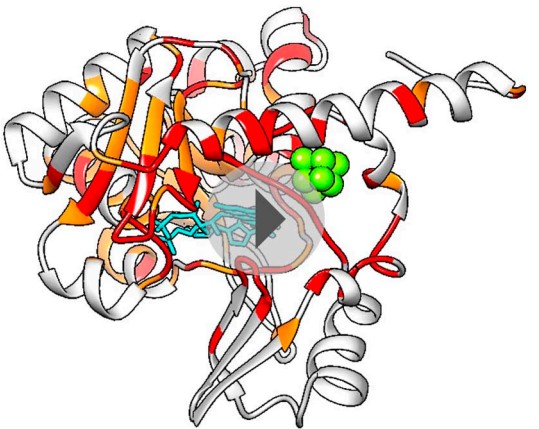

**Video 6**. Conserved residues in FrhB. The ligands FAD and FeS are shown in magenta and green, respectively. Conserved residues and corresponding densities are shown in red and partially conserved residues in orange. Both ligands and the $F_{420}$ access channel are completely surrounded by conserved residues.

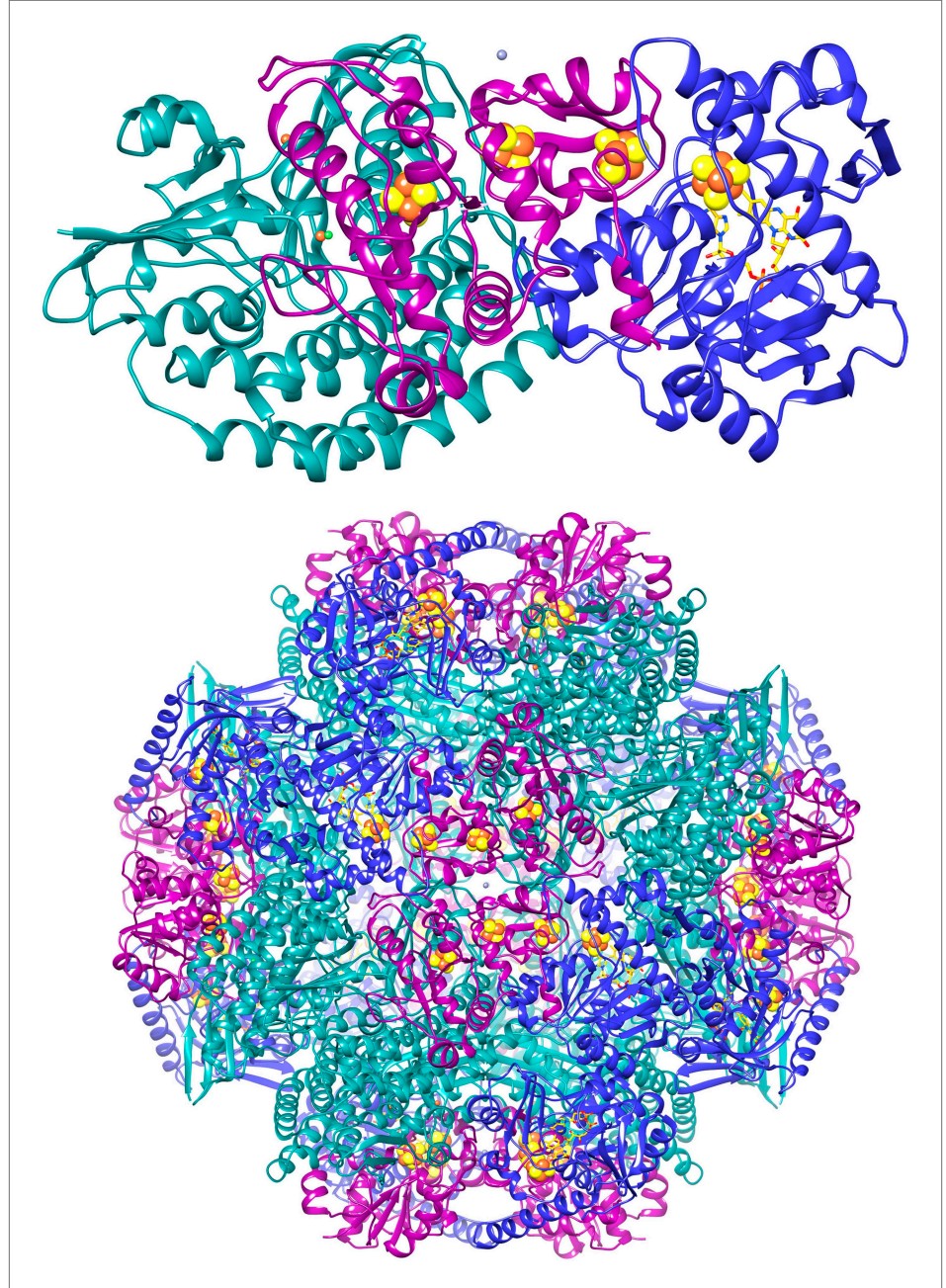

**Figure 14**. Cartoon of the FrhABG heterotrimer (top) and the tetrahedral complex of 12 trimers. FrhA is green, FrhG magenta, and FrhB blue. The [NiFe] center in FrhA is shown as green and orange spheres, the three [4Fe4S] clusters as orange and yellow spheres, and the FAD in FrhB as a stick model with yellow carbons, the ion in FrhA is orange, and the putative zinc ion on the twofold axis of the FrhG dimer is grey.

calibrated magnification on the 14 μm pixel camera was 106,000, resulting in a 1.32 Å pixel size at the specimen. The camera system was set up to record 18 frames/sec as previously described (*Bai et al., 2013*). Videos were collected for 1.5 s with a total of 24 frames with a calibrated dose of 3.5 e⁻/Å² per frame, at various defocus values in the range between 0.8 and 3.8 μm.

## Image processing

Particle picking was carried out using the semi-automatic procedure of EMAN Boxer (*Ludtke et al., 1999*), and the contrast transfer function of every image was determined using CTFFIND3

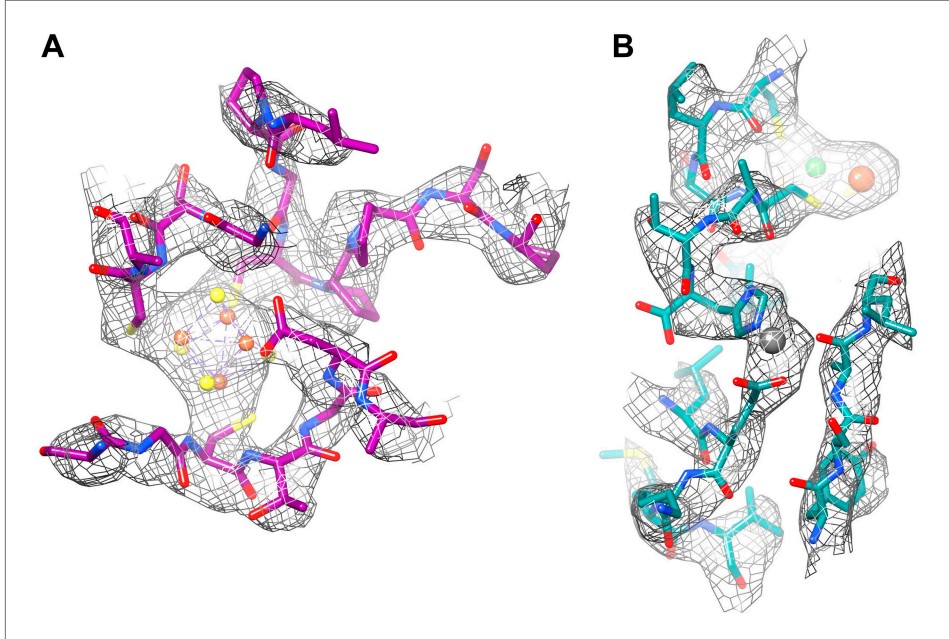

**Figure 15**. (**A**) The proximal FeS cluster of FrhG is coordinated by three cysteine residues and an aspartate (Asp60) with clear density. (**B**) An ion in FrhA (grey sphere) is coordinated by the C-terminal His386, the main chain oxygen of Ala347 and by Glu44, one of the few carboxylate residues with clear density.

(*Mindell and Grigorieff, 2003*) in the RELION workflow (*Scheres, 2012*). If necessary, the CTF values were double-checked using the particle-based CTF procedure of EMAN2 (*Tang et al., 2007*). Four independent refinements were launched with the gold standard refinement procedure of RELION (*Scheres and Chen, 2012*) starting from our previous Frh map from film data (*Mills et al., 2013*) low-pass filtered to 60 Å, using 20 frames (from frame 2 to frame 21), testing four different approaches:

1. The 20 unaligned frames of each video were added up without motion correction;
2. The 20 frames of each video were aligned using the whole-image motion correction method described in *Li et al., 2013* (the authors kindly provided the necessary support to install the software);
3. The 20 unaligned frames were processed by the statistical video refinement procedure described in *Bai et al. (2013)* (particle-based);
4. A combination of approaches 2 and 3, applying the statistical video refinement procedure (*Bai et al., 2013*) to the 20 frames pre-aligned using the whole-image motion correction software (*Li et al., 2013*).

Procedure number 4, which combines the area-based and particle-based frame alignment, gave the best results. All further maps as described in the Results section were obtained by RELION refinement as in procedure 4, unless less than 8 video frames were used, in which case procedure 2 was followed. Tetrahedral symmetry was applied in all refinements.

A post-processing procedure implemented in RELION (*Scheres, 2012*) was applied to the final maps for appropriate masking, B-factor sharpening and resolution validation to avoid over-fitting (*Chen et al., 2013*). In this procedure, the appropriate B-factor is determined according to *Rosenthal and Henderson (2003)*, after correction for the Falcon II MTF. In addition, a soft mask is applied to the last two unfiltered models before convergence and a new FSC curve is calculated. The procedure also measures any spurious correlation due to too tight masking, by subtracting the FSC curve between the two masked half datasets where the phases beyond a chosen resolution were randomized. For all maps, B factors between −150 and −230 Å$^2$ were found. All resolutions stated are at FSC 0.143 (*Rosenthal and Henderson, 2003*) after applying this post-processing procedure.

The local resolution of the map was estimated with the ResMap software (available at http://resmap.sourceforge.net) (*Kucukelbir et al., 2014*).

## Model building

The protein structure was built in Coot (*Emsley and Cowtan, 2004*) into the high-resolution map using real-space refinement, starting with the earlier model from cryo-EM data recorded on film (*Mills et al., 2013*). Torsion angle, planar peptide, and Ramachandran restraints were applied throughout. The final model for the FrhABG heterotrimer contains 893 amino acid residues out of a possible 903, a [NiFe] cluster, four [4Fe4S] clusters, one FAD, and two metal ions, one of them coordinated by two trimers. 96.5% of residues have backbone dihedral angles in the most favored region of the Ramachandran plot and the remaining 3.5% are in the generously allowed regions.

Figures were made using Chimera (*Pettersen et al., 2004*).

## Acknowledgements

We thank Stella Vitt and Seigo Shima for providing the protein, Xueming Li for help in installing his frame alignment software, and Karen Davies for determining the MTF of the Falcon II detector. We further thank Juan Castillo and Özkan Yildiz for computer support and Edoardo D'Imprima and Susann Kaltwasser for discussions and help in picking particles.

## Additional information

### Competing interests

WK: Reviewing editor, *eLife*. The other authors declare that no competing interests exist.

### Funding

| Funder | Author |
|---|---|
| Max Planck Society | Werner Kühlbrandt |
| UK Medical Research Council | Greg McMullan |

The funders had no role in study design, data collection and interpretation, or the decision to submit the work for publication.

### Author contributions

MA, Acquisition of data, Analysis and interpretation of data, Drafting or revising the article; DJM, Acquisition of data, Drafting or revising the article; GM, Acquisition of data, Drafting or revising the article, Contributed unpublished essential data or reagents; WK, Conception and design, Drafting or revising the article; JV, Conception and design, Analysis and interpretation of data, Drafting or revising the article

## Additional files

### Major datasets

The following datasets were generated:

| Author(s) | Year | Dataset title | Dataset ID and/or URL | Database, license, and accessibility information |
|---|---|---|---|---|
| Allegretti M, Mills DJ, McMullan G, Kühlbrandt W, Vonck J | 2013 | Electron cryo-microscopy of F420-reducing [NiFe] hydrogenase Frh | EMD-2513; http://www.ebi.ac.uk/pdbe-srv/emsearch/atlas/2513_summary.html | In the public domain at the Electron Microscopy Data Bank (EMDB) at PDBE: http://www.ebi.ac.uk/pdbe/emdb/. |
| Allegretti M, Mills DJ, McMullan G, Kühlbrandt W, Vonck J | 2013 | Atomic model of the F420-reducing [NiFe] hydrogenase by electron cryo-microscopy using a direct electron detector | 4CI0; http://www.rcsb.org/pdb/search/structidSearch.do?structureId=4ci0 | Publicly available at the RCSB Protein Data Bank (http://www.rcsb.org/pdb/). |

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
