## [Decision Letter]

Thank you for sending your work entitled “Near-atomic model of the F_420_-reducing [NiFe] hydrogenase by cryo-electron microscopy using a direct electron detector” for consideration at *eLife*. Your article has been favorably evaluated by a Senior editor, John Kuriyan, and 3 reviewers, one of whom, Wesley Sundquist, is a member of our Board of Reviewing Editors.

The Reviewing editor and the other reviewers discussed their comments before we reached this decision, and the Reviewing editor has assembled the following comments to help you prepare a revised submission.

The authors describe a high resolution electron cryomicroscopy (cryo-EM) reconstruction of the F_420_ reducing NiFe hydrogenase from the methanogenic archaeon, Methanothermobacter marburgensis (Frh). Frh is an exceptionally well behaved, heterotrimeric enzyme with overall Td symmetry, providing 12-fold molecular averaging. The new 3.36 Å resolution reconstruction represents a significant improvement on the ∼5 Å reconstruction reported last year by the same group.

The strengths of the study are that it provides: 1) a reconstruction that is “state-of-the-art” in terms of methodology and resolution, 2) helpful comparative analyses of different methodological approaches, and 3) additional information on the Frh structure, including side chain positions. The authors found that the reconstruction resolution was optimized when they a) averaged frames 2-7 (omitting the first frame owing to large sample motions and omitting subsequent frames owing to radiation damage), b) omitted images with the thicker ice, high defocus values and poor Thon rings, and c) combined the whole frame and particle alignment methods. With the optimized reconstruction, the authors were able to: a) confirm that the previous structure was correct, b) assign two new surface helices, c) identify a new Zn^2+^(Cys)4 site at the FhrG dimer interface, d) argue that the enzyme’s active site does not appear to adjust substantially upon F_420_ substrate binding, e) add to the available data concerning electron beam-induced decarboxylation of Asp and Glu residues (most such side chains are decarboxylated, but several are not, including several that coordinate cofactors), and f) demonstrate that reliable near atomic-resolution information was present even in their previous EM maps that exhibited nominal overall ∼5 Å resolution, thanks to the direct phase information present in the images.

The study is of high technical quality and it provides another excellent example of the ongoing revolution in cryo-EM, where improved detector quantum efficiencies are increasing image signal-to-noise and improved software packages and data analysis approaches are making better use of the improved signal. The manuscript is as important for its demonstration of technical capability as for its biological insights.

Issues that should be addressed prior to publication:

1) Since the authors interpret their density in terms of an atomic model (and they deposit coordinates) they should include in the paper, in addition to all the FSC curves shown in Figures 2 and 3, at least one FSC between their best map and their best model. If their interpretation of the density map in terms of the model is mostly correct, then the map/model FSC should have a value of 0.5 at the same resolution where the gold-standard FSC between maps calculated from the two independent halves of the experimental data is 0.143. If it is not roughly in that range, then they should try to improve their model. This is a very rigorous test and should be included in any paper where a map and model are presented.

2) In Figure 2, and the legend that describes it; the authors may have misunderstood the procedure in [9] that is required to determine the extent of overfitting. In that paper, there were two uses of the idea of randomizing the phases. In the first, all the raw single particle images have their phases randomized beyond a chosen resolution. The entire structure determination is then repeated and equation 4 from Chen et al. is then used to calculate the true FSC in which any overfitting can be measured and removed to produce an FSCtrue plot. However, there is also another use of equation 4 of Chen et al shown in their Figure 4, which was used to determine the effect of any aggressive masking. In this second application of equation 4, the final 3D map has its phases randomized beyond a chosen resolution, and the FSC and the corrected FSCtrue is calculated to measure and remove the effect of masking. From the figure legend in Allegrini et al., it is unclear whether the authors have carried out the first or the second procedure. The first does indeed test for overfitting, whereas the second test only for over-aggressive masking. If the authors have done only the second, then all they have really done is to test whether the Relion masking is soft. In that case, they should carry out the (more time-consuming) procedure of repeating the entire structure determination using randomized high-resolution phases for the raw single particle images. Failing that, if the FSC test proposed in item 1 ends up being ∼0.5, (between the best density and the best model), then it is probably sufficient simply to correct the Figure 4 legend regarding the use of the HR-noise test (and remove all references to determination of overfitting, assuming they haven’t actually done that).

3) The issue of substrate occupancy should be clarified. It appears that the authors have made a typographical error in the Discussion when they say that: “We used an Frh sample with a large excess of FAD to calculate...” (presumably they mean to say “F_420_” rather than “FAD”?). The more significant issue is that it is unclear why they previously assigned density for the isoalloxazine ring of F_420_ in the F_420_-replete sample but now don’t see any density in the new higher resolution map. Looking back at the previous F_420_ density, it appears that this could well have been noise, but if this was the case then it should be stated clearly (or, if not, the discrepancy should be explained). The authors should also exercise care in how they phrase the conclusion that they expect no conformational changes when F_420_ binds if they don’t really have a structure of the F_420_-bound enzyme.

---

## [Author Response]

*1) Since the authors interpret their density in terms of an atomic model (and they deposit coordinates) they should include in the paper, in addition to all the FSC curves shown in*
Figures 2 and 3*, at least one FSC between their best map and their best model. If their interpretation of the density map in terms of the model is mostly correct, then the map/model FSC should have a value of 0.5 at the same resolution where the gold-standard FSC between maps calculated from the two independent halves of the experimental data is 0.143. If it is not roughly in that range, then they should try to improve their model. This is a very rigorous test and should be included in any paper where a map and model are presented*.

We added this curve as Figure 10. The map versus model FSC is 0.5 at a resolution of 3.56 Å, close to the gold-standard map resolution of 3.36 Å.

*2) In*
Figure 2*, and the legend that describes it; the authors may have misunderstood the procedure in*
[9]
*that is required to determine the extent of overfitting. In that paper, there were two uses of the idea of randomizing the phases. In the first, all the raw single particle images have their phases randomized beyond a chosen resolution. The entire structure determination is then repeated and*
*equation 4*
*from Chen et al. is then used to calculate the true FSC in which any overfitting can be measured and removed to produce an FSCtrue plot. However, there is also another use of*
*equation 4*
*of Chen et al shown in their*
Figure 4*, which was used to determine the effect of any aggressive masking. In this second application of*
*equation 4**, the final 3D map has its phases randomized beyond a chosen resolution, and the FSC and the corrected FSCtrue is calculated to measure and remove the effect of masking. From the figure legend in Allegrini et al., it is unclear whether the authors have carried out the first or the second procedure. The first does indeed test for overfitting, whereas the second test only for over-aggressive masking. If the authors have done only the second, then all they have really done is to test whether the Relion masking is soft. In that case, they should carry out the (more time-consuming) procedure of repeating the entire structure determination using randomized high-resolution phases for the raw single particle images. Failing that, if the FSC test proposed in item 1 ends up being ∼0.5, (between the best density and the best model), then it is probably sufficient simply to correct the*
Figure 4
*legend regarding the use of the HR-noise test (and remove all references to determination of overfitting, assuming they haven’t actually done that)*.

The reviewers are correct; we had in fact performed the second test that measures the effect of the masking. The aim of Figure 2 was not to check for overfitting, which was prevented by our use of the Relion refinement procedure, but to show the resolution of the map after masking. We have thus corrected the legend.

*3) The issue of substrate occupancy should be clarified. It appears that the authors have made a typographical error in the Discussion when they say that: “We used an Frh sample with a large excess of FAD to calculate...” (presumably they mean to say “F*_*420*_*” rather than “FAD”?). The more significant issue is that it is unclear why they previously assigned density for the isoalloxazine ring of F*_*420*_
*in the F*_*420*_*-replete sample but now don’t see any density in the new higher resolution map. Looking back at the previous F*_*420*_
*density, it appears that this could well have been noise, but if this was the case then it should be stated clearly (or, if not, the discrepancy should be explained). The authors should also exercise care in how they phrase the conclusion that they expect no conformational changes when F*_*420*_
*binds if they don’t really have a structure of the F*_*420*_*-bound enzyme*.

We corrected the error and changed “FAD” to “F_420_”. We assume that the density found in the film map was due to noise and indicated this in the manuscript. We also rephrased the conclusion.